# Biostimulants as an Alternative to Improve the Wine Quality from *Vitis vinifera* (cv. Tempranillo) in La Rioja

**DOI:** 10.3390/plants11121594

**Published:** 2022-06-16

**Authors:** Cristina E. Olavarrieta, Maria Carmen Sampedro, Asier Vallejo, Nikola Štefelová, Ramón J. Barrio, Nuria De Diego

**Affiliations:** 1Department of Analytical Chemistry, Faculty of Pharmacy, University of the Basque Country UPV/EHU, 01006 Vitoria-Gasteiz, Spain; asier.vallejo@ehu.eus (A.V.); r.barrio@ehu.eus (R.J.B.); 2Central Service of Analysis (SGIker), University of the Basque Country UPV/EHU, Lascaray Ikergunea, Miguel de Unamuno 3, 01006 Vitoria-Gasteiz, Spain; mariacarmen.sampedro@ehu.eus; 3Centre of Region Haná for Biotechnological and Agricultural Research, Czech Advanced Technology and Research Institute, Palacky University, Šlechtitelů 27, 78371 Olomouc, Czech Republic; nikola.stefelova@seznam.cz

**Keywords:** *Vitis vinifera* L., biostimulants, water deficit, grapevine, growth stages, phenology, primary metabolism

## Abstract

The application of biostimulants appears to be an environmentally friendly, innovative, and sustainable agronomical tool to mitigate the negative effects induced by adverse climatology in traditional grape-growing regions such as La Rioja (Spain). However, their mechanism of action in grapevines is still unclear. We evaluated how commercial substances (two from *Ascophyllum nodosum* extraction and one amino acids-based biostimulant) and the non-proteinogenic amino acid β-aminobutyric acid (BABA) affect the quality and quantity of musts and grapes in *Vitis vinifera* L. cv. Tempranillo from a semi-arid region of La Rioja during two seasons. We hypothesized an enhancement in organic metabolites in berries and leaves in response to these treatments, changing the organoleptic characteristics of the final products. The treatments altered the primary metabolites such as carbohydrates, organic acids (AcOrg), and free amino acids, first in the leaves as the effect of the foliar application and second in grapes and musts. As the main result, the biostimulant efficiency depended on the climatology and vineyard location to improve the final yield. Whereas biostimulant application enhanced the yield in 2018 (less dry year), it did not help production in 2019 (dry year). BABA was the most efficient biostimulant, enhancing plant production. Regarding yield quality, the biostimulant application improved the musts mainly by enhancing the fumaric acid content and by reducing carbohydrates, except in BABA-treated plants, where they were accumulated. These results corroborate biostimulants as an exciting approach in wine production, especially for improving wine quality.

## 1. Introduction

The wine-making sector represents some of the most important social, cultural, and economic activities in many regions, especially in Spain. According to the International Organization of Vine and Wine [1] report, Spain has the highest vine growing extent destined for wine production, with 966 mha and a production of 33.5 Mill. hL (12.9% of the world total). Among the 97 production regions in Spain, DOCa Rioja is considered the region par excellence for wine production due to the climatic diversity. In 2019, a 2.7 Mill. hL was registered by “Consejo Regulador de la denominación de origen Rioja” [2].

La Rioja is a semi-arid region where the Mediterranean climate prevails, with long, hot, and dry summers [3], for which the wine-growing develops under limited water conditions. Although the grapevine plants are drought tolerant, environmental consequences related to climate change negatively impact vineyard growth and productivity [4,5]. Some studies showed that the increase in temperature and drought intensity due to climate change reduced yield and berry quality [6,7]. Other studies reported changes in phenological events and growing seasons [3,8,9,10].

Drier seasons end with a higher alcoholic content in wines [8]. This is mainly due to a higher accumulation of sugar, which conditions the acidity of the wine [5,11,12,13]. However, this accumulation depends on the level of stress and the period it appears [14,15,16]. Generally, the vine is more sensitive to water deficits after flowering and fruit formation, although additional water supplies can reverse this negative impact [17,18,19]. However, in Europe and in La Rioja, vineyards are non-irrigated crops. Furthermore, Ramos and Martinez-Casasnovas (2010) [20] indicated that when the water deficit during the growth period is higher than 50% of crop evapotranspiration (ET_c_), the yield can also be affected.

Other agricultural techniques such as green water use [21] and the use of genetic manipulation [22], including vineyard migration toward higher latitudes [23,24,25], could reduce the adverse effects of environmental stresses and increase the resistance and resilience of crops [26,27]. However, these techniques are costly and require a total change of the still productive grapevine plants. In the last year, growers have increased their interest in using more environmentally friendly solutions. The application of biostimulants is one of these solutions, and the demand for this kind of product is still increasing. They are used in agriculture to enhance the productivity and quality of plants and to increase the plants’ resilience to stress due to their sustainable properties [28,29,30]. Biostimulants comprise a broad type of formulations, including complex compositions based on humic and fulvic acids, protein hydrolysates and other N-containing compounds, seaweed extracts, biotic and abiotic elicitors, and beneficial fungi and bacteria [28,31]. Seaweed-based biostimulants are among the most studied substances [32,33]. For example, the use of biostimulants based on common brown seaweeds such as *Ascophyllum nodosum* can improve plant productivity and quality, photosynthesis [34,35], root development [36,37], and drought, salinity, and freezing tolerance in *Vitis vinifera* L. [38,39,40]. These biostimulants are complex and contain many organic components, including free amino acids (AAs), proteins, vitamins, polysaccharides, and plant hormones [33,35]. However, due to their complexity, the mechanism of action that their application activates in plants is still unclear, especially in crops such as grapevine.

It is well known that stress conditions modify plant metabolism, which also happens in grapevines. In plants, free AAs, carbohydrates, and AcOrgs are compatible solutes involved in osmotic regulation and stress tolerance [41,42]. However, prolonged stress can induce the accumulation of free amino acids (AAs) [43,44] and the premature depletion of organic acids (AcOrg) [45,46,47]. AcOrgs have several functions in plant metabolism [48], increasing their biosynthesis, accumulation, transport, and root exudation in response to stressful environmental conditions [49]. In wine, the content of AcOrg affects the final product’s stability, color, and flavor, but their most significant role comes from their ability to stop, or at least retard, the growth of many potentially harmful microorganisms that would spoil the wine itself [50].

AAs are essential for wine quality; thus, they are the main grape N compounds, the principal source of N for yeasts, and precursors of several volatile compounds [51]. Some of them are also considered to be stress-related compounds. For example, γ-aminobutyric acid (GABA) is rapidly accumulated in plants under stress to improve tolerance [52]. Furthermore, its exogenous applications help deal with stress in many crops [52,53,54]. Foliar pulverizations of β-aminobutyric acid (BABA), an isomer of GABA, enhance the ability to quickly reply to water stress and improve the resistance to many diseases such as mildew [55,56]. In this work, we investigated the effect of three commercial biostimulants, two from *Ascophyllum nodosum* extracts and another one rich in amino acids due to its involvement in plant stress tolerance. Additionally, the application of BABA was included as an example of a small molecule-based biostimulant and as a positive control to evaluate the mechanism of action of more complex substances. We hypothesized that biostimulant application improves plant performance and the yield quantity and quality under water deficit.

## 2. Results

### 2.1. The Two Seasons Differed in Climatology and Water Availability for the Vineyard

The precipitation and temperature of the growing season were followed to understand the climate’s impact on our vineyard. Figure 1a shows that higher precipitations occurred in 2018 during the budbreak but then reduced until maturity. Contrarily, T_Max_ and T_min_ increased. These climatic conditions induced a high increase in the effective crop evapotranspiration (ETc), especially at bloom and to lesser extent at veraison. Consequently, the vineyard presented lower water deficit (WD) in these two developmental stages and reached the lowest values at maturity (Figure 1b).

A different profile was observed in 2019. This year was dried, and the precipitations were lower than in 2018 for all development stages analyzed, especially at veraison (V) (Figure 1c). Only higher precipitations were registered at maturity. Additionally, at bloom, T_Max_ reached 40.5 °C, which was 7 °C more than in 2018 (Figure 1c). A low pluviometry at veraison and high temperatures at bloom increased ETc and reduced the WD in the vineyard, showing that the plants suffered a higher restriction of water available for the crop than in 2018 during the relevant developmental stages.

### 2.2. Biostimulants Modified the Water Balance of the Plants

Before and after the second application, the plants’ water balance, calculated by leaf water content (LWC, %) was studied to clarify the ambient and application effects (Figure 2). The application was performed during bloom as specified in material and methods.

Before the application, LWC was lower in the plants in 2019 than in 2018 (Figure 2a), probably due to the high temperatures recorded (Figure 1b). The foliar application negatively impacted the LWC, reducing the values in both seasons, and only those plants sprayed with Greetnal and Basofoliar improved their LWC in 2018 and 2019, respectively (Figure 2a). Subplots A and B were then analyzed separately to better understand the biostimulant effect on LWC. The ratio between the LWC of the plants after (T24B) and before (T0B) the second foliar application was represented for better visualization of the results (Figure 2b).

Again, an apparent negative impact of the foliar application in the LWC was visible in all the treatments in 2018, including the controls (Figure 2b). The LWC was lower in those plants from subplot A than from B. The same happened in 2019. However, in this case, only the plants sprayed with Basofoliar increased the LWC at T24B compared to the same plants at T0B in both A and B subplots (Figure 2b). The plants sprayed with Greetnal tended to maintain or increase the LWC in A or B, respectively (Figure 2b). Altogether, the results showed that a second application with biostimulants could positively affect the water balance of the wine plants when applied after bloom, especially in the hotter year.

### 2.3. Biostimulant Applications Improved the Production under Moderate but Not Severe Stress Conditions

At the end of each season, different parameters related to yield and final plant survival (%) were determined in the vineyard. Again, apparent differences were observed between subplots and years and the interaction of both (Figure 3).

No variations in the yield-related parameters were observed between treated and untreated plants in 2019 (Figure 3). However, their foliar application modified the number of clusters and the dimensions of the grapes in 2018. For example, biostimulants reduced the clusters per plant in 2018, except for BABA and Basofoliar or SoilExpert in subplots A and B, respectively (Figure 3a).

BABA application enhanced the weight of the grapes and the final vine yield (kg) in subplot A but not in B, compared to controls (Figure 3b,e). One possible reason for the good results observed with BABA was a better fruit setting and the low incidence of diseases in the treated plants (Appendix A). Contrarily, Basofoliar and SoilExpert increased the grape diameter compared to untreated vines in subplot A in 2018.

The vine survival (%) in the old and low productive vineyards was also determined because the study was performed. No considerable effect on plant survival was observed (Figure 3f). The best results were observed when BABA was applied in subplots B in 2018 and A in 2019. Contrarily, the SoilExpert application reduced survival in subplots A in 2018 and B in 2019 (Figure 3f). Altogether, we can conclude that the efficiency of the biostimulants strongly depends on the interactive effect between the year (climatology) and the characteristics of the plot and that the non-protein amino acid BABA is the most promising biostimulant for plant production and survival.

### 2.4. Biostimulant Applications Modified Water Use Efficiency Related to Production

The plant’s water use efficiency (WUE) was calculated as a next step. As a result, we observed that only the plants sprayed with BABA significantly increased the WUE compared to the controls in subplot A in 2018 (Figure 4). The commercial substances did not change or decrease WUE (Figure 4). Only the application of Greetnal slightly improved the WUE in subplot B in 2018 without significant differences (Figure 4).

### 2.5. A Better Physiological Response Supports a Higher Vine Yield

To better visualize and simplify biostimulant characterization, we used the Plant Biostimulant Characterization (PBC) index developed by Ugena et al. (2018) [57]. This approach ends with a positive or negative value that permits the classification of the biostimulant. This index is calculated by the sum of all values represented in a parallel coordinate plot per variant. This value is the log2 of the ratio between the untreated and treated plants [57].

Four parallel coordinate plots were performed, two for the PBC index representing plant physiology and survival for the 2018 and 2019 seasons, and two additional ones with the production-related parameters (Figure 5). As the main result, an adverse effect of the biostimulant application was observed in the vines located in subplot B (discontinuous lines, Figure 5), which ended with negative values of PBC index (Table 1).

Only the use of GT improved plant physiology in 2018 (Table 1). In subplot A, the foliar application with biostimulants improved vine physiology and yield in 2018 but not in 2019; thus, all variants presented positive PBC index values. No correlation was observed between the PCB index values for physiology and yield. However, the variants with positive values of PBC index for physiology (mainly better water balance) ended with better yield or maintained it. Regarding survival, BABA was the most effective treatment, as it improved the survival in 2018 and did not change in 2019 compared to the control plants (Figure 5a,b). Altogether, we can conclude that the application of biostimulants on vines improves final yield when they can induce a more efficient use of the water in the plant, which can help to survive under certain growth conditions.

### 2.6. The Application of Biostimulants Altered the Metabolite Content in Vines

The content of concrete-free AAs [=(L-aspartic acid (Asp), L-glutamic acid (Glu), L-asparagine (Asn), L-glutamine (Gln), L-arginine (Arg), L-alanine (Ala), GABA, BABA and α-aminobutyric acid (AABA)), carbohydrates (D-(-)-fructose (Fruct), D-(+)-glucose (Gluc), sucrose (Sac), and maltose (Malt)), and AcOrg (oxalic acid (Oxal), L-(+)-tartaric acid (Tart), DL-malic acid (Mal), acetic acid (Acet), citric acid (Cit), succinic acid (Suc), and fumaric acid (Fum)) was performed in different matrices (leaves, grapes, and musts) (Figure 6 and Figure 7 and Appendix A). Carbohydrates were the most abundant metabolites in all samples, followed by organic acids (g kg^−1^ dry weight). Free amino acids and the organic acid Fum appeared at lower concentrations (mg kg^−1^ dry weight) (Appendix A). Due to the complex experimental setup, we performed a multivariate statistical analysis based on partial least squares discrimination analysis (PLS-DA) for better visualization. This approach permitted the direct comparison between control and treatment, ending with four PLS-DA figures for each type of matrix. The distribution obtained by the PLS component 1 and 2 separated the effect by season (2018 or 2019) and sampling time in leaves. Only the application of BABA showed an apparent effect in the leaves; they accumulated Gluc, Fruct, and Mal, whereas the control plants accumulated more Sac and Cit (Figure 6a). The biggest influence was the season in the rest of the treatments, with leaves accumulating Gluc, Fruct, BABA, and GABA in 2019 or Fum, Mal and Asp in the latest harvesting times in 2018.

In grapes and must, the differences between treatments were more evident, but the content of metabolites among both matrices also differed (Figure 7, Appendix A). In grapes, the PLS distribution was mainly due to the season. For example, the grapes for all the plants accumulated Tart, Acet, and Gln in 2019. The content of Fruct and Gluc was more related to the 2018 season, except for BABA plants that accumulated higher levels in the grapes in 2019. This accumulation was only partially visible in musts from BABA-treated vines, where Fruct and Sac were also related to Fum and Suc. The Basofoliar application also induced the accumulation of Fum together with GABA in musts, independent of the year and subplot. When Greetnal or SoilExpert were used, different metabolite profiles in must were observed. These plants mainly accumulated Fruct and Gluc when they were treated and located in subplot A. Altogether, the foliar application with biostimulants affected the metabolite profile of the musts more than of the grapes. However, each substance altered the composition differently, with AcOrg and sugars as those with more relevant changes.

### 2.7. The Commercial Biostimulants Varied in Their Metabolic Composition

As the final step, we performed a principal component analysis to compare the content of the studied metabolites in the commercial products used. Again, different profiles were observed, including between Greetnal and SoilExpert, biostimulants based on the seaweed *Ascophyllum nodosum*. Whereas Greetnal presented the highest Gln and Acet values, SoilExpert contained more Fum, Tart, and Oxal (Table 2, Figure 8). Basofoliar presented the highest amount of free AAs and the organic acid, Mal (Figure 8). The differences in the metabolic profiles could also explain the specificities observed in the plant response.

## 3. Discussion

The use of biostimulants in grapevines is one of the most significant interests of wine producers to reduce the use of chemicals that negatively affect plant and human health [58,59,60]. They reported that biostimulants based on simple molecules or more complex substances could help the plant deal with biotic and abiotic stresses. Biostimulants have been described to improve stress resilience mainly by increasing photosynthesis, modifying metabolism, and activating the antioxidative response of the plants. The biostimulants based on *Ascophyllum nodosum* are not an exception, as described by [39,60,61,62]. *Ascophyllum nodosum*-based biostimulants can improve plant performance and the yield and quality of plants subjected to different stress conditions [61,63]. We wanted to further characterize the biostimulant mechanism of action in our work. We compared the use of three commercial substances, one amino acid-rich formulation, two *Ascophyllum nodosum* extracts, and the non-proteinogenic amino acid BABA as a small molecule based-biostimulant on an old vineyard highly affected by climate change. In general, the climatology of the year and the vine plant location (subplot) highly condition the effectiveness of the biostimulant application, especially in the water use efficiency and yield-related parameters. In this regard, it has been reported that the position of the subplot and especially the row orientation could condition physiological functioning [64] and the metabolic profile [65]. In our study, the physiology-related parameters and yield quantity were more affected in A than B. The row orientation is the same for both subplots. However, they presented different adjacent objects. Whereas A is close to the main road, B is located between the other plots with other vineyards. This makes subplot A more susceptible to pollution and other stresses and could partially explain the differences. The differences in the metabolic profile, at least for leaves and grapes, were mainly due to the season.

As mentioned above, the application of biostimulants in vineyards has been related to better plant physiology under stress conditions. In our work, the biostimulant-treated plants also improved the LWC and WUE related to final production, most probably via the content of stress-related metabolites such as GABA, BABA, Mal and Cit, among others (Figure 2 and Figure 4, Table 2). However, the location and the year highly influenced the plant response. Whereas biostimulants enhanced LWC during the driest season (2019), WUE was improved only in subplot A in 2018. Using the PBC index to compare the results related to plant physiology and yield (Figure 5), we can conclude that WUE but not the LWC condition plant production; thus, only the biostimulant-treated vine plants from subplot A in 2018 presented positive values. One possible explanation is that the high temperatures are the most limiting factor in grapevine [66]. We observed that the maximal temperatures in 2019 during the relevant phenological stages such as bloom and veraison were much higher than in 2018. BABA was the most efficient biostimulant, improving vine yield mainly by berries with higher diameter. One of the reasons could be the low incidences of disease observed in the BABA-treated plants in 2018 (Appendix A and Figure 9). BABA has been well documented as a predominant priming elicitor for effective resistance induction and is desirable for disease management in agricultural fields [56,67]. Its application has also been used to induce resistance in grapevine (*Vitis vinifera*) against downy mildew [55] or to increase the biotic resistance in postharvest grapes [68]. The positive effect of BABA in production was not visible in subplot B in 2018. However, in this subplot, its application increased the most regarding the survival of the plants (Figure 5a). Kocsis et al. [69] observed that BABA can also inhibit flower fertility and hence berries per cluster and weight, pointing to the interaction between BABA treatment and the environmental pressure as the determining factor for the positive or negative response of the plant. BABA application also increased the accumulation of Glu, Fruct and Mal in leaves. All are acting as compatible solutes that allow the plant to maintain the water status under stress conditions [42,70] and hence to survive better under stress. Finally, WUE has been described as the capacity of a crop to produce biomass per unit of water evapotranspiration and hence the major component of yield [71]. We could conclude that biostimulant application can enhance vine plant production, but the positive effect depends on the climatological conditions and the plant location. Furthermore, the result obtained regarding WUE pointed to this physiological parameter as an excellent biomarker to predict vineyard yield.

The most studied parameters for defining the quality of the berries and must in vine plants treated with biostimulants are anthocyanins and phenolic compounds [61,72]. However, in this study, we focused on the content of sugars, free AAs, and AcOrg. In the first conclusion, we could see that the application of biostimulants modified the metabolite distribution; thus, differences were observed between the profiles of grapes and must. In grapes, multivariate statistical analysis showed that the changes in carbohydrates, AAs, and AcOrg were mainly due to the growing season. In musts, the effect of the biostimulant applications was more evident with AcOrg and the carbohydrates Gluc and Fruct as the main contributors to the differences. It has been shown that the increase in temperatures in Europe, North America, and Australia affects grape composition, inducing an increment in the sugar concentration and a decrease in acidity, ending with poor-quality wines [73]. In this context, we observed that the application of biostimulants reduced the sugar content mainly in must, especially when the commercial substances were applied. However, in grapes, the sugar content varied among treatments. Whereas in BABA-treated plants, Fruct and Gluc accumulated in 2019, for the rest of the treatments, accumulation occurred in 2018. The accumulation in 2018 could be mainly due to a delay in veraison and maturity and hence grape ripening. However, BABA treatment allowed the plants to perform better than the rest of the treatments and control in 2018, but not in 2019, mainly due to an efficient effect of the plant diseases control.

The application of Basofoliar and BABA enhanced the content of Fum. Wine freshness depends significantly on its acidity, and Fum has been reported to inhibit malolactic fermentation or stop it once initiated to preserve the wine’s Mal content. However, the increase in Fum in BABA-treated plants was most correlated with higher Fruct, Gluc and Suc. Altogether, we think that the accumulation of sugars and AcOrg could compensate for the alcohol degree and acidity.

Basofoliar-treated plants also increased GABA. Amino acids are considered relevant in wines because they are precursors for aroma compounds and directly contribute to the wine’s aroma, taste, and appearance [74]. GABA can serve as a source of nitrogen to produce Suc, and its assimilation improves yeast growth, fermentation rate, and glycerol production [75]. However, too much Suc induces a salty–bitter taste in the wine [76]. Additionally, wine rich in GABA has a scientific interest because it has been demonstrated that GABA possesses many physiological functions such as regulation of blood pressure, heart rate, and hormone levels, as well as reduction of blood lipid and improvement in liver and kidney function in humans [77]. Altogether, it is clear that the biostimulant application modified the composition of the must and hence conditions the final product and wine characteristics.

## 4. Materials and Methods

### 4.1. Grapevine Field and Experimental Design

The field experiments were carried out in a red Tempranillo (*Vitis Vinifera* L.) vineyard located in Cenicero, La Rioja (Spain) (42°27′ N, 2°39′ W and altitude at 436 m above sea level) over two growing seasons (2018 and 2019). The vineyard was planted in 1989 in a north–south orientation. Grapevines were grafted onto 110-Richter rootstock and grown on clay and sandy soil at a planting density of 3365 vines ha^−1^. The plants were trained to a regular system with nine rows in rainfed conditions and grown using cultural practices that DOCa Rioja provides.

The experimental design was a split-plot with two subplots (A facing northwest and B at southeast orientation, Figure 10). In each subplot, three commercial biostimulants were applied using foliar application following the producers’ recommendations to improve the quality and quantity of the grapevine production; Basofoliar Avant Natur (BF, Martinez Carra S.L., Calahorra, La Rioja) based on 100% plant AAs extracts, applied at 3 L ha^−1^, and two additional products with a similar origin, SoilExpert (ST, Alfa & Omega Consulting S.L., Puente La Reina, Navarra) and Greetnal (GT, Lainco S.A., Barcelona), both based on seaweed *Ascophyllum nodosum* extracts and applied at 2.5 L ha^−1^ (Figure 1). Two additional treatments were included: a negative control (C) using water at the same volume and a positive control applying the non-protein amino acid BABA at a concentration of 0.1 mM (Figure 1).

The treatment was performed in all the vines per line and subplot. However, due to the dimensions of the experiment, five vines (one every three strains) with similar developmental stages per row and plot were selected for the study of plant water balance, production and metabolomics. High or low topographic positions and vigorous or weak strains were avoided to compare more homogeneous samples. All samples were collected in the early hours of the day to prevent enzymatic degradation of the plant material.

For the analysis of the leaves, five different samplings were made; before and 24 h after the first foliar application defined as T0A and T4A, before and 24 h after the second foliar application defined as T0B and T24B, and after harvest (TF). Six leaves per plant were collected during each sampling time. Young leaves were used in the early stages of growth (located on the same side of the first bunch), and at veraison, leaves opposite the first bunch were used because they are abundant and provided more information about plant reserves because they support the production of fruit.

For grapes and must, an additional two samplings were performed; before (September, TV) and after the harvest (October, TF) when the fruit was fully ripened. Before the harvest, the basal and consecutive bunches were collected. The other two bunches were also harvested from the remaining brunches during the ripening. Two homogeneous grapes per plant were removed from the bunches collected. They were homogenized, and two aliquots were made for each sample. The first aliquot was squeezed using a mortar and filtered to obtain the pulp juice. The second aliquot was crushed with their seeds and skins to obtain a grape paste. All samples were stored at −40 °C until analysis.

### 4.2. Climate Data

During the 2018 and 2019 seasons, the climatic conditions were recorded in meteorological stations placed nearby the analyzed plots in Cenicero. The information included daily maximum temperature, daily average temperature, daily minimum temperature, and precipitation. The average temperature and precipitation referred to the growing season (1st March–30th September) were calculated from these data. The average dates for budbreak, bloom, veraison, and maturity recorded during the analyzed period were considered to define the different periods between phenological stages. The starting point of the growth season (budbreak) was determined by direct observation and daily monitoring of the vineyard.

In addition, the reference crop evapotranspiration (ET_0_, mm) was estimated as proposed by Hargreaves and Samani (1985) [78] in the following equation:(1)$${\mathrm{ET}}_{0}=\left(0.0023\right)\cdot \left(T_{\mathrm{avg}}+17.8\right)\cdot {\left(T_{\mathrm{Max}}-T_{\min}\right)}^{0.5}\cdot R_{a}$$
where R_a_ is the extraterrestrial solar radiation (mm day^−1^).

Then, ET_c_ (mm) was also determined. This term corrects the deficiencies of the previous equation by a Kc factor that depends on the moisture soil level, crop characteristics, and the stage of the crop vegetative cycle (Equation (2)) [79]. For vines whose fruits are destined for wine production and with a maximum height between 1.5–2 m, K_c_ values are estimated at 0.3 in the initial phases of the cycle, 0.7 in the intermediate phases, and 0.45 in the final stages of the vegetative cycle [79].
(2)$${\mathrm{ET}}_{c}={\mathrm{ET}}_{0}\cdot K_{c}$$

Finally, WD were quantified as accumulated precipitation minus crop evapotranspiration. Water deficits were determined during growing seasons (1 March–30 September); budbreak, bloom, veraison, and maturity for periods between phenological stages as defined by [10]. The accumulated ET_c_ from budbreak to maturity was also calculated and used to estimate the WUE (kg mm^−1^), which is the ratio between the accumulated ET_c_ and the final yield (kg ha^−1^).

### 4.3. Leaf Water Content

LWC (%) was calculated in leaves after the second foliar application and was determined as:(3)$$\mathrm{LWC}\;\left(\%\right)=\frac{\mathrm{FW}-\mathrm{DW}}{\mathrm{DW}}*100$$
where FW is the fresh weight of leaves, and DW is the dry weight of the leaves after 24 h of drying in the oven at 60 °C.

### 4.4. Yield Related Parameters and Plant Survival

For all the variants, yield parameters were evaluated at harvest by assessing vine yield (kg), cluster number per vine (n), average cluster length (cm), average cluster weight (g), and grape diameter (cm) in both studied seasons. After the dormancy, the vine’s survival (%) for each variant was also estimated counting all the plants per variant.

### 4.5. Free Amino Acids Quantification by HPLC-FLD

Three independent pools as biological replicates were used to measure the content of certain free AAs for the three evaluated types of samples (leaves, grape, and must). For leaves, the extraction of free AAs was performed as described by De Diego et al. (2013) [42], with minor modifications. Then, 0.1 g of lyophilized material containing 5 µL of 2-aminoadipic acid (as internal standard) was extracted, and the mixture was mixed for 10 min and centrifuged at 10,000 rpm for 10 min at 4 °C. Pellets re-extracted an additional 1 mL of ethanol. Supernatants were collected and diluted ten times before injection. All samples were diluted with the mobile phase at initial conditions and filtered through 0.22 µm polyvinylidene fluoride (PVDF) syringe filters before injections. The extraction of free AAs in the grape paste was performed as Kelly et al. (2010) [80] described. Then, 0.2 g of samples with 20 µL of the internal standard was extracted with 980 µL of ethanol. The supernatant was collected and diluted at 1:50 (*v*/*v*). The must was analyzed by direct injection of 2 mL of must containing 10 µL of the internal standard according to Gutiérrez-Gamboa et al. (2018) [63]. Then, samples were diluted at 1:20 (*v*/*v*) before the injection and filtered through 0.22 µm PVDF syringe filters.

All extracted samples were quantified with the same procedure as described below. Free AAs: Asp, Glu, Asn, Gln, Arg, Ala, GABA and BABA from Alfa Aesar (Karlsruhe, Germany) were quantified according to the method described by Agilent Technologies (2017) [81]. The analytical method was carried out on an 1100 Agilent HPLC equipped with a quaternary 1260 Agilent technologies pump and an Agilent 1260 fluorescence detector (FLD), all from Agilent Technologies (Santa Clara, SA, USA) using 9-fluorenylmethyl chloroformate (FMOC) as derivatization agent. Twenty microliters of each mixture were injected onto an InfinityLab Poroshell 120 HPH-C18 (2.7 µm, 100 × 4.6 mm, Agilent Technologies, Santa Clara, SA, USA) with a guard column InfinityLab Poroshell 120 (Analytical Guard Column 2.7 µm, 5 × 4.6 mm, ©Agilent Technologies, Inc., Palo Alto, CA, USA) at 40 °C. Spectra were obtained using the DataAnalysis program for HPLC-FLD (Agilent Technologies, Inc., Palo Alto, CA, USA).

### 4.6. Organic Acids Quantification by HPLC-DAD

The AcOrgs (Oxal, Tart, Mal, Acet, Cit, Suc, and Fum) were extracted as Hazer et al. (2016) [82] described, with minor modifications. For leaves, 20 mL of NaOH (0.1mM) was added to 5 g of fresh material, and the mixture was kept at 4 °C for 24 h. Then, samples were placed in an ultrasonic bath for 30 min and filtered. The mixture was washed with 5 mL of NaCl (5% *w*/*v*) and shaken for 5 min. Finally, samples were rinsed twice in 1.5 mL of Millipore water and were acidified with H_3_PO_4_ to pH 2.7. The extraction of the organic acids from the grape paste was performed as described by Flores et al. (2012) [83]. Then, 3 g of each sample homogenized with 10 mL of H_2_O was centrifuged at 10,000 rpm for 5 min at 4 °C. Supernatants were diluted at 1:2 (*v*/*v*) and acidified with H_3_PO_4_ to pH 2.7. In must, homogenized samples were centrifuged at 10,000 rpm at 4 °C for 15 min and diluted at 1:3 as described by Zheng et al. (2009) [84]. All samples were diluted with the mobile phase at initial conditions and filtered through 0.22 µm PVDF syringe filters before injection.

For quantification, an HPLC system consisted of an Agilent HPLC Model HP 1100 Series (Agilent Technologies, Santa Clara, SA, USA). The chromatographic separation was performed in an analytical UltraAqueous C-18 analytical column (150 cm × 4.6 mm i.d. × 5 μm particle size) from Waters Corporation (Milford, MA, USA) and thermostated at 25 °C.

The mobile phases were (A) 99:1 NaH_2_PO_4_/ACN solution (4mM, pH 2.7, with H_3_PO_4_) and (B) ACN. First, 20 μL of the sample was injected and analyzed in a mobile phase gradient with a flow rate of 0.5 mL min^−1^. The elution was programmed with 0% B for 4.9 min, 0–20% B for 7.1 min, 20–60% B for 1 min, 60% B for 2 min, 60–0% B for 2.5 min, and 0% B for 2.5 min followed by column re-equilibration in A for 5 min. Photodiode Array detection was set a 210 nm. ChemStation Rev. B.04.02 software (Agilent Technologies©, Palo Alto, CA, USA) was used for data acquisition, peak integration, and standard calibration

### 4.7. Carbohydrates Quantified in Berries and Leaves by HPLC-RID

Carbohydrates (Fruct, Gluc, Sac and Malt) were extracted from leaf samples as previously described by Lunn et al. (2006) [85] and Luo et al. (2019) [86]. First, 250 µL of ice-cold chloroform blended with methanol (3:7, % *v*/*v*) was added to 10 mg of dry weight samples and vortex for 30 s. Then, samples were incubated at −20 °C for 2 h with occasional vortex mixing. Next, 200 µL of ice-cold water was added to the mixture, and the samples were vortex for 3 min until they were tempered.

Then, the samples were centrifuged (10,000 rpm) at 4 °C for 10 min using a 5415R Eppendorf™ microcentrifuge from Marshall Scientific (Hampton, NH, USA). The aqueous-methanol phase was transferred to new 1.5 mL centrifuge tubes. As described above, the chloroform phase was re-extracted with 200 µL of ice-cold water. The aqueous-methanol phases were mixed and evaporated to dryness under a gentle nitrogen stream. The dry residues were reconstituted in 250 µL of ACN/H_2_O (80:20). Finally, all the samples were centrifuged at 8000 rpm for 5 min and filtered through 0.22 µm PVDF syringe filters.

For carbohydrate determination in grape samples, 200 µL of ice-cold water was added to 100 mg of the homogenized material and centrifuged at 10,000 rpm at 4 °C for 10 min. The extracts were diluted at 1:20 in ACN/H_2_O (80:20) and centrifuged at 8000 rpm for 5 min. In musts, 980 μL of ice-cold water was added to 20 µL of homogenized samples. Then, the mixtures were shaken for 5 min and centrifuged at 10,000 rpm at 4 °C for 10 min. All measurements were filtered through 0.22 µm PVDF before the injection.

HILIC-RID analyses were carried out on an Agilent 1100 series HPLC system supplied with Agilent 1100 series quaternary pump and Agilent 1260 degasser coupled to a refractive index detector (RID) Agilent 1260 from Agilent Technologies (Santa Clara, CA, USA) operating at 45 °C. Chromatographic separation was performed on a ZIC^®^-HILIC SeQuant^®^ (3.5 µm, 150 × 4.6 mm i.d.; Merck, Darmstadt, Germany) protected with ZIC^®^-HILIC SeQuant^®^ guard column (5 µm, 20 × 2.1 mm i.d.; Merck, Darmstadt, Germany) keeping at 60 °C during the run. The flow rate was 0.7 mL min^−1^, and the injection volume for each sample was 5 µL. The mobile phase was composed of ACN/H_2_O (80:20, % *v*/*v*), and the LC run time was 20 min using an isocratic elution, followed by column re-equilibration of 5 min. Agilent LC ChemStation software (Agilent Technologies, Santa Clara, CA, USA) was used for system control and data analysis.

### 4.8. Statistical Analysis

Data analysis was performed using RStudio (R Software version 4.1.0, https://cran.r-project.org/bin/windows/base/) using packages, agricolae, ggplot2, pls. Data were normalized for the subsequent analysis. Three-way ANOVAs, including treatment (control, BABA, GT, BF or ST), subplot (A or B), and year (2018 or 2019) as factors followed by Duncan’s tests were used to define the differences in the yield-related parameters. For the metabolic data, multiple four-way ANOVAs including treatment (control vs. compound), time (sampling time), plot, and the year (2018 or 2019) as factors followed by Duncan’s tests for multiple comparisons were performed, using Bonferroni corrections for multiple testing. PLS-DA was performed using the SIMPLS algorithm and taking treatment (control vs. one of the others) as the dependent (binary) variable. PLS biplots were displayed.

## 5. Conclusions

Biostimulants have been proven valuable for improving wine quality from vineyards located in areas highly affected by climate change. They can enhance the content of organic acids and specific amino acids (i.e., GABA) and reduce sugars, improving the quality of the final product. However, not all biostimulants work the same way, including when the raw material source is the same. It is, then, essential to characterize them before their general use. Moreover, the application of biostimulants can also change the physiology and the final yield, but in this case, the climatology and location of the vineyard also influence the plant response. Finally, we demonstrated that water use efficiency (WUE) related to production can be a useful biomarker to predict the yield of the vineyard and hence the plant biostimulant efficiency.

## Data Availability

Not applicable.

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
