# Peer review of "Biostimulants as an Alternative to Improve the Wine Quality from Vitis vinifera (cv. Tempranillo) in La Rioja"

_plants, 2022, doi:10.3390/plants11121594_

Round 1

Reviewer 1 Report

The manuscript might be of interest to Plants readers, but it must be reviewed very carefully. 

The term climate change appears in the title, but I think it is not appropriate given the focus of the study conducted.

The abstract needs to be improved by more accurately detailing the results obtained, perhaps reducing the introductory part.

The scope of the paper in the final part of the introduction needs to be revised as it is confusing and the objectives of the research need to be better clarified.

line 136: It does not appear that the Greetnal had a significant effect, compared to the control, on LWC (Fig. 2A).

140-141: The effect of biostimulants must be defined as positive or negative with reference to the control, and not arbitrarily.

188-189: Check the significance indices as they do not appear to be in agreement with Fig. 4

189: Greetnal does not appear to be statistically different from the control, as indicated in the sentence. 

Figures 6 and 7 need to be improved as they are difficult to read. 

233-234: A table showing the quantitative data of all molecules studied in this work is needed.

The discussion should be revised by trying to give an explanation for the results obtained.

Furthermore, how do the authors explain the many variabilities between subplots A and B in?

Author Response

The manuscript might be of interest to Plants readers, but it must be reviewed very carefully. 

First of all, we would like to thank the reviewers for their comments and suggestions to improve our manuscript. We hope that the current version is suitable to be published in Plants.

The term climate change appears in the title, but I think it is not appropriate given the focus of the study conducted.

The title was modified, removing the climate change appearance. Besides, the text was checked and corrected according to the reviewer's suggestions.

The abstract needs to be improved by more accurately detailing the results obtained, perhaps reducing the introductory part.

The introductory part of the abstract was reduced, and the results expanded as suggested by the reviewer.

The scope of the paper in the final part of the introduction needs to be revised as it is confusing and the objectives of the research need to be better clarified.

The final part of the introduction was revised and modified. Additionally, the working hypothesis was also included to clarify better the aim of the study.

line 136: It does not appear that the Greetnal had a significant effect, compared to the control, on LWC (Fig. 2A).

There is a mistake that was corrected in this sentence. Greetnal and Basofoliar improved their LWC in 2018 and 2019, respectively (Fig. 2A). The mistake has been corrected  (Line 133)

140-141: The effect of biostimulants must be defined as positive or negative with reference to the control, and not arbitrarily.

The lines 140-141 were clarified. The comparison with each other performed in Fig. 2B was mainly to analyze the effect of the second foliar application. As a result, we observed that in some treatments in 2019, the second application increased the LWC, when it was expected a reduction because of the treatment effect.

188-189: Check the significance indices as they do not appear to be in agreement with Fig. 4

The text in these two lines was corrected and clarified according to the results observed in Fig. 4

189: Greetnal does not appear to be statistically different from the control, as indicated in the sentence. 

The information that there are no significant differences was included at the end of the paragraph (Line 187)

Figures 6 and 7 need to be improved as they are difficult to read. 

Figures 6 and 7 have been revised and improved the quality to be more visible as suggested by the reviewer

233-234: A table showing the quantitative data of all molecules studied in this work is needed.

We included the table with the average numbers and standard error of each metabolite for each variant (New Tables 2 and 3). This will help the readers to see the values of each tissue.

The discussion should be revised by trying to give an explanation for the results obtained.

A big part of the discussion has been modified according to the reviewers' comments and the changes made in other sections.

Furthermore, how do the authors explain the many variabilities between subplots A and B in?

We also included the differences between subplots in the first paragraph of the discussion section (Lines 309- 316). We concluded that the differences are due to the adjacent objects. Whereas A is close to the main road, B is located between other plots with other vineyards.

Reviewer 2 Report

The paper deals with a topic of considerable interest and current relevance.

Unfortunately, the experimental design appears not very consistent, and the treatments are carried out only on 5 vines for each plot, without further repetitions.

This can lead to a high variability of the sample, which affects the responses to treatments

While interesting results are reported on some analytical aspects, for example in relation to the effect on the amino acid content, other important parameters are missing in reference to the physiological responses of the vines, such as the water potentials of the leaves or the photosynthetic activity.

In general, the terminology used is not always adequate (e.g. line 57 the term rainfed crops is used instead of non-irrigated, or the term wine grape is used instead of vinegrape).

Graphs with PCA are often used, but tables with variance analysis of individual measured values would be more useful.

In detail, the following aspects and corrections are noted.

Introduction:

Line 60: What is the meaning of "green roots"? Perhaps the authors meant "green water use"?

Line 69: Replace fluvic acid with fulvic acid.

Results:
Line 119-120: Reference to Figure are incorrect.

Line 124: Figure 1 highlights meteorological parameters useful for describing the environmental conditions in which the research took place. They can be moved to the materials and methods chapter.

All the figures show the phenological phases, but it is not clear to which time intervals they refer.

In Figures 1c and 1d the values of ETc and WD are obviously inverted.

Line 132: Reference to figure 8 is not relevant to the context of the sentence.

Line 134-135: Authors should try to better explain this result.

Line 159-160: It is impossible for the treatments to modify the number of clusters per vine in the first vintage (the differentiation to flower occurs in the previous year!). Perhaps this is due to the low number of plants sampled?

Figure 3: In 2018, the control has an extremely low weight of the bunches, and at the same time shows a considerable average length: how can it be explained?

Even the often lower % of survival in treated vines compared to control is poorly explained.

Line 185: The authors cite a parameter (WEU) that usually refers to the photosynthetic efficiency of plants and not to the ratio between ETc and production/ha.

Materials and Methods:

Authors should better detail the differences between subplots A and B - are they different exposures or topographic positions of the vineyard?

Scheme 1 is superfluous, it is advisable to replace it with a table with phenological phases and dates of the treatments.

Line 339: Replace ripped with ripened.

The water content of the leaves was measured as the ratio between fresh weight and dry weight, but this parameter has little significance from a physiological efficiency point of view. Measuring the leaf or stem water potential would be much more useful.

Line 442: For the amino acid analysis were only biological replicas made and not between plots?

The method of analysis of A.A.s it's too detailed. If it is not original, it is sufficient to cite some references in bibliography.

Conclusion:

The reduction of sugars in grapes is mentioned, but it is not highlighted whether it depends on reduced drying due to less water loss or for an effective delay in ripening.

Author Response

The paper deals with a topic of considerable interest and current relevance.

First of all, we would like to thank the reviewers for their comments and suggestions to improve our manuscript. We hope that the current version is suitable to be published in Plants.

Unfortunately, the experimental design appears not very consistent, and the treatments are carried out only on 5 vines for each plot, without further repetitions.

We want to inform the reviewer that the treatment was applied to all the plants for each line. We only selected 5 vines per variant for the analysis due to the dimensions of the experiment. It means 5 plants* 5 treatments * 2 subplots= 50 samples. We collected leaves four times = 200 samples. The grapes and must were collected at 2 additional times = 100 for grapes and 100 for must. This is a total of 400 samples per year. Considering that we measured 3 types of compounds, we go to more than 2000 samples, plus the time for quantification of each metabolite and data analysis. This is not easy to manage on a bigger scale. We should include the measurements for production-related parameters for all this work. Only for the survival were used all the plants per variant.

This can lead to a high variability of the sample, which affects the responses to treatments

As shown in our data, the variability among samples was more due to the year (2018 was more variable) than the position of the plants. This was also visible in the metabolite analysis because, in some cases, the differences were mainly due to the year. We agreed that a higher number of individuals per variant can reduce it but, as mentioned above, it was not possible due to the experiment dimensions.

While interesting results are reported on some analytical aspects, for example in relation to the effect on the amino acid content, other important parameters are missing in reference to the physiological responses of the vines, such as the water potentials of the leaves or the photosynthetic activity.

We agreed with the reviewer that a deeper physiological approach could answer some additional questions. However, this study was more focused on the study of the biostimulant effect on the production, and more importantly on the quality of the final product.

In general, the terminology used is not always adequate (e.g. line 57 the term rainfed crops is used instead of non-irrigated, or the term wine grape is used instead of vinegrape).

The terminology was rechecked and corrected. We also changed the term rainfed by non-irrigated (line 59) or wine grapes by vinegrape (Line 21).

Graphs with PCA are often used, but tables with variance analysis of individual measured values would be more useful.

We used PCAs because under multidimensional studies they allow us to understand better the results. They can simplify the data and remark the most valuable information. This makes the complex stories easier to understand and discuss. Our statistician also performed a four-way ANOVA in which all parameters showed significant interaction for the four factors (treatment, subplot, year, and time) (see attachment). However, we didn't include this information because the reading will be boring and tedious. Additionally, from these data is very difficult to extract conclusions.

In detail, the following aspects and corrections are noted.

Introduction:

Line 60: What is the meaning of "green roots"? Perhaps the authors meant "green water use"?

Green Root Wines. Made and bottled with a respect for nature, to taste good at a fair price and leave the smallest possible carbon footprint.

In the line 62 we included green water use.

Line 69: Replace fluvic acid with fulvic acid.

The correction was done in line 72

Results:
Line 119-120: Reference to Figure are incorrect.

We corrected the references in this paragraph according to Figure 1.

Line 124: Figure 1 highlights meteorological parameters useful for describing the environmental conditions in which the research took place. They can be moved to the materials and methods chapter.

We left the meteorological data together with the ETc and Wd because they are used for the calculation and together will make easier the understanding of the changes to the readers.

All the figures show the phenological phases, but it is not clear to which time intervals they refer.

The intervals and time per year were now included in the scheme representing all experiments (Scheme 1), this way; the readers can follow the information about the phenological phases better.

In Figures 1c and 1d the values of ETc and WD are obviously inverted.

They are not inverted. They are calculated as shown in the material and methods. For better clarification, we also include an excel file with the calculations to allow the reviewer to check (See supplementary file for the reviewer).

Line 132: Reference to figure 8 is not relevant to the context of the sentence.

The reference of Figure 8 was relocated to a sentence that is related to it.

Line 134-135: Authors should try to better explain this result.

We think that the biostimulants helped the LWC due to the content of stress-related metabolites such as GABA, BABA, malate, and citric acid, among others. It can be observed in Table 4. This was also included in the discussion (Lines 345-346).

Line 159-160: It is impossible for the treatments to modify the number of clusters per vine in the first vintage (the differentiation to flower occurs in the previous year!). Perhaps this is due to the low number of plants sampled?

2018 was a more wet year full of storms. As mentioned in the text, this influenced the production, and many plants from specific treatments suffered a high fungus infection (Supplementary material). This conditioned the number of clusters per vine. This difference was not evident in 2019 because the year was drier, and the plants didn't suffer so much from the fungus attacks.

Figure 3: In 2018, the control has an extremely low weight of the bunches, and at the same time shows a considerable average length: how can it be explained?

As mentioned above, in 2018 some of the grapes were highly infected by fungus so the phenotype of the grapes was modified and sometimes was not round.

Even the often lower % of survival in treated vines compared to control is poorly explained.

The results (Lines 363-370) and discussion (Lines 365-366 and 371-72) sections include a text regarding the plant survival.

Line 185: The authors cite a parameter (WEU) that usually refers to the photosynthetic efficiency of plants and not to the ratio between ETc and production/ha.

The water use efficiency is a physiological parameter, usually calculated by gas exchange parameters. In this case, there are two; instantaneous WUE, the net photosynthesis/transpiration, or the intrinsic WUE calculated by net photosynthesis/stomatal conductance. However, the water use efficiency can be defined as the amount of carbon assimilated as biomass or grain produced per unit of water used by the crop (Haldiend and Dold, 2019, https://doi.org/10.3389/fpls.2019.00103). In this case, there are several examples in the literature that calculate the relation between the yield and the total ET to understand the plant's water use per production. For example:

Puppala et al 2005. https://doi.org/10.1016/j.indcrop.2003.12.005

Hao et al 2019. https://doi.org/10.1007/s00271-018-0597-5

Materials and Methods:

Authors should better detail the differences between subplots A and B - are they different exposures or topographic positions of the vineyard?

We also included the differences between subplots in the first paragraph of the discussion section (Lines 309- 316). We concluded that the differences are due to the adjacent objects. Whereas A is close to the main road, B is located between other plots with other vineyards.

Scheme 1 is superfluous, it is advisable to replace it with a table with phenological phases and dates of the treatments.

We improved the scheme 1, including the phenological phases and dates of the treatments

Line 339: Replace ripped with ripened.

Ripped was changed by ripened (Line 456)

The water content of the leaves was measured as the ratio between fresh weight and dry weight, but this parameter has little significance from a physiological efficiency point of view. Measuring the leaf or stem water potential would be much more useful.

We agreed that the leaf or stem water potential analysis could be a beneficial physiological parameter to define the degree of stress between plants. However, as mentioned above, the aim of the work was more metabolic profiling than a physiological study.

Line 442: For the amino acid analysis were only biological replicas made and not between plots?

All the analyses for all metabolites were done using biological replicates per treatment, subplot, year, and harvesting time.

The method of analysis of A.A.s it's too detailed. If it is not original, it is sufficient to cite some references in bibliography.

We reduced the text related to the amino acid quantification as suggested.

Conclusion:

The reduction of sugars in grapes is mentioned, but it is not highlighted whether it depends on reduced drying due to less water loss or for an effective delay in ripening.

A discussion part about sugar accumulation has been included in lines 393-398.

Round 2

Reviewer 1 Report

The authors have answered all the questions and the paper has significantly improved. Now it can be accepted for publication.

Reviewer 2 Report

The authors have made appreciable but not sufficient improvements to the text.

Understanding that further modifications (in particular to the experimental design and to the parameters detected) are not feasible, considering the quality of "Plants", it is proposed not to accept the paper for publication.

The same can be proposed for other journals of the MDPI publisher.